# Influence of Lateral Sitting Wedges on the Rasterstereographically Measured Scoliosis Angle in Patients Aged 10–18 Years with Adolescent Idiopathic Scoliosis

**DOI:** 10.3390/bioengineering10091086

**Published:** 2023-09-14

**Authors:** Andreas Feustel, Jürgen Konradi, Claudia Wolf, Janine Huthwelker, Ruben Westphal, Daniel Chow, Christian Hülstrunk, Philipp Drees, Ulrich Betz

**Affiliations:** 1Department of Orthopaedics and Trauma Surgery, University Medical Centre of the Johannes Gutenberg University Mainz, D-55131 Mainz, Germany; 2Institute of Physical Therapy, Prevention and Rehabilitation, University Medical Centre of the Johannes Gutenberg University Mainz, D-55131 Mainz, Germany; 3Institute of Medical Biostatistics, Epidemiology and Informatics, University Medical Centre of the Johannes Gutenberg University Mainz, D-55118 Mainz, Germany; 4Department of Health & Physical Education of The Education University of Hong Kong, Hong Kong; 5Asklepios Katharina-Schroth-Klinik Bad Sobernheim, D-55566 Bad Sobernheim, Germany

**Keywords:** spinal deformity, adolescent idiopathic scoliosis, surface topography, scoliosis angle, sitting, seat wedge, lateral inclination

## Abstract

Adolescent idiopathic scoliosis (AIS) is a three-dimensional axial deviation of the spine diagnosed in adolescence. Despite a long daily sitting duration, there are no studies on whether scoliosis can be positively influenced by sitting on a seat wedge. For the prospective study, 99 patients with AIS were measured with the DIERS formetric III 4D average, in a standing position, on a level seat and with three differently inclined seat wedges (3°, 6° and 9°). The rasterstereographic parameters ‘scoliosis angle’ and ‘lateral deviation RMS’ were analysed. The side (ipsilateral/contralateral) on which the optimal correcting wedge was located in relation to the lumbar/thoraco-lumbar convexity was investigated. It was found that the greatest possible correction of scoliosis occurred with a clustering in wedges with an elevation on the ipsilateral side of the convexity. This clustering was significantly different from a uniform distribution (*p* < 0.001; chi-square = 35.697 (scoliosis angle); chi-square = 54.727 (lateral deviation RMS)). It should be taken into account that the effect of lateral seat wedges differs for individual types of scoliosis and degrees of severity. The possibility of having a positive effect on scoliosis while sitting holds great potential, which is worth investigating in follow-up studies.

## 1. Introduction

Adolescent idiopathic scoliosis (AIS) is a three-dimensional spinal axis deviation [1]. It is the most common growth deformity, with a prevalence of 2–3% [2] and is manifested by asymmetries such as a rib hump or pelvic obliquity [1]. Other symptoms may include psychological impairment and a reduction in quality of life due to the disease itself or therapy [2,3]. The aetiology is not yet fully understood, but it is thought to be multifactorial [2]. The diagnosis of AIS is based on history, clinical examination and radiological imaging of the entire spine in two planes. The curve pattern and the Cobb angle are determined [4]. Scoliosis can be classified according to severity and topography. Depending on the degree of the most pronounced lateral curvature, a distinction is made between thoracic, thoraco-lumbar, lumbar and combined (s-form; with similar degrees of curvature on two levels) scolioses. The severity is determined by the extent of the Cobb angle. There are low (Cobb < 20°), moderate (Cobb < 40°) and severe (Cobb > 40°) scolioses [2]. Therapy depends on the severity, the age of the patient and the associated residual growth and ranges from observational waiting to scoliosis-specific physiotherapy, corset therapy and surgical treatment [2]. Low and moderate scolioses are the most common forms, which is why conservative therapy, both in the form of physiotherapy and by means of a corset, is the primary treatment for many patients [2]. It has been shown that this prevents the progression of scoliosis [5] and even reductions in the Cobb angle have been described [6,7].

Frequent X-rays are performed for follow-up in patients with scoliosis, which increases the risk of developing cancer [8,9,10]. For this reason, alternative non-radiological measurements of the spine have been developed. These include systems that use light-optical measurements to calculate a three-dimensional spine model based on surface topography. A light grid is projected onto the patient’s back and the distortions of the light grid through the surface are recorded with a camera [11]. This creates a ‘virtual plaster cast’ [12] of the back, from which in turn conclusions can be drawn about the underlying position of the spine [12]. These ‘videorasterstereographic’ measurements enable a radiation-free representation of the spine and are thus suitable for repeated measurements.

As the ‘central organ of movement’, the spine is both dynamic and flexible [4]. Due to the close relationship between the pelvis and the spine—Dubousset even speaks of the ‘pelvic vertebra’ [13]—the spine reacts to changes in the pelvic position. This is also true for patients with scoliosis, despite the structural nature of the scoliotic curve. In some patients, especially those with a difference in leg length, the scoliosis can also be positively influenced by shoe lifts [14,15]. However, this intervention can have an effect only when the patient is standing or walking. As children and adolescents between the ages of 4 and 20 spend an average of 9.7 h per day sitting, which corresponds to about 70% of the time spent awake [16], the investigation of possible similar effects due to sitting on laterally inclined wedges is of great interest.

For this reason, this study measured the influence of laterally inclined seating surfaces through seat wedges on the spine for the first time using rasterstereography. The aim was to find out on which side related to the thoraco-lumbar/lumbar convexity the elevation of a seat wedge should be in order to achieve an optimal correction of scoliosis. Secondary investigations were also conducted to determine whether the degree of inclination required for optimal correction depends on the severity and type of scoliosis and whether a more steeply inclined wedge produces a better correction. In addition, the subjective sitting sensation was surveyed to determine whether a measured correction is also accompanied by an improvement in the sitting sensation.

## 2. Materials and Methods

Between September 2020 and February 2021, patients of the Katharina-Schroth-Klinik Bad Sobernheim were informed by mail about the study, informed verbally during the admission interview and included in the study if they gave their consent and met the inclusion criteria. The inclusion criteria were:Patients with an age of 10–18 years (including 18);A Cobb angle of 10–50° with lumbar/thoraco-lumbar involvement;No pain (numeric rating scale (NRS) ≤ 4);No acute illness, no chronic diseases with influence on balance;No previous surgery;BMI < 30 kg/m^2^;No change in the surface of the back due to large scars/tattoos.

Verbal and written consent was obtained from all patients and their legal guardians. The Ethics Committee of the Rhineland-Palatinate Medical Association gave a positive vote for the study (application number: 2020-15047).

### 2.1. Method

The DIERS formetric III 4D average system was used for the measurements in this study. This is a video rasterstereographic method in which the surface topography of the back is recorded and the position of the spine is calculated consecutively with the help of automatically identified points (VP: vertebra prominens (usually corresponds to cervical vertebra 7); DL/DR: dimple left/dimple right) and calculated points (e.g., DM: dimple middle; middle between DL and DR) in their three-dimensionality. This method has been tested many times for validity, reliability and reproducibility [12,17,18,19,20,21,22,23]. Even if rasterstereography cannot replace X-ray diagnostics, it is nevertheless suitable for reliably revealing changes in the spine [18]. The accuracy of the system has been subject of several studies. The automatic localization detects the landmarks in standing subjects with a deviation of 1 mm compared to radiological control [17,24]. The rastersterographic scoliosis angle deviates from the Cobb angle by 7–8° [25]; there was found to be a high and significant correlation (r > 0.7; *p* < 0.0001) [19].

For the present study, patients were measured in a standing position, in a sitting position on a flat seat surface and on laterally inclined seat wedges made of acrylic glass with 3°, 6° and 9° inclination. The elevation of the seat wedges was on the contralateral and ipsilateral side of the thoraco-lumbar/lumbar convexity of the scoliosis in each case (partly demonstrated in Figure 1). Thus, a total of eight measurements were taken per patient. The order of the sitting wedges was randomized. For better detection of the lumbar dimples, especially in the sitting position, they were marked with reflective markers. The patients were told to sit upright and let their arms hang loosely at the sides. Before each measurement, the patient had a 30 s familiarization period. After this time, the patient was asked about their sitting sensation, and then the measurement was started.

### 2.2. Target Parameters

The parameters ‘scoliosis angle’ and ‘lateral deviation RMS’ (RMS: root mean square) were analysed. The rasterstereographic scoliosis angle is calculated automatically and describes the angle between the tangents at the vertebral bodies most inclined towards each other (the rotation of the vertebral bodies is mathematically integrated into the extent of the tilt). The lateral deviation RMS is defined as the root mean square deviation of the distance between the spine line and the VP–DM line. The optimal correction was defined as the intra-individual minimum of the scoliosis angle parameter or the lateral deviation RMS parameter. The sitting sensation was recorded by means of a numerical rating scale. This ranged from 0 (very comfortable) to 10 (very uncomfortable).

### 2.3. Statistics

The collected data were recorded, analysed and exported using the software DICAM V.3.12.2 from the company DIERS. The program IBM SPSS Statistics V.27 was used for statistical analysis. A chi-square test was applied to test the observed distribution against an expected uniform distribution. The level of significance was set to *p* < 0.05.

## 3. Results

Initially, 128 patients aged 10–18 years with AIS were included. At the time of measurement, they were at the beginning of a rehabilitation stay at the Katharina-Schroth-Klinik Bad Sobernheim. Subsequently, 29 patients were excluded from the evaluation. The reasons for this were subsequently differently classified scoliosis (severity, type, age) (*n* = 24), discontinuation at the patient’s request (*n* = 1), language barrier (*n* = 1), pain (NRS ≥ 4) on the day of measurement (*n* = 1, the measurement was nevertheless performed at the patient’s request) and faulty or missing individual measurement (*n* = 2). Finally, 99 patients, with a sex ratio of 8:1 (female = 88; male = 11) were analysed. Table 1 shows the epidemiological data. Table 2 shows the distribution of scoliosis types and severity. Figure 2 shows that the distribution of the degrees of severity within the individual types of scoliosis is different. Whereas thoracic scolioses are more often severe, about 85% of lumbar scolioses are low or moderate.

It was shown that for both the scoliosis angle and the lateral deviation RMS, the wedge with the inclination on the ipsilateral side led more often to optimal correction (see Figure 3): 61.6% of the patients achieved optimal correction of the scoliosis angle and 67.7% that of the lateral deviation RMS by a wedge with an ipsilateral elevation. The distribution deviated significantly from the uniform distribution (scoliosis angle: Pearson chi-square = 35.697; df = 2; *p* < 0.001/lateral deviation RMS: Pearson chi-square = 54.727; df = 2; *p* < 0.001).

When considering the exact distribution of optimally correcting wedges according to the severity and scoliosis type, it can be seen that in lumbar scolioses the optimal correction is more often achieved by a wedge with ipsilateral elevation, whereas in thoracic scolioses no accumulation is evident. There is a trend that the more caudal the main curve of the scoliosis, the more the clustering of optimal correction shifts to the side of the ipsilateral wedge elevation. The exact distribution of optimally correcting wedges is shown in Figure 4. Each point represents one patient, grouped by severity and scoliosis type, and indicates at which inclination the optimal correction occurred.

Looking at the change in the mean value of each parameter in the seat compared to the optimal correction, it could be seen that the amount of improvement was different. By subgrouping according to scoliosis type and severity, it could be shown that especially lumbar scolioses and those with smaller Cobb angles were better corrected. Lumbar scolioses could be corrected by up to 51% (lateral deviation RMS); low scoliosis showed a correction of approximately 36% (lat. deviation RMS). This trend was evident for both the scoliosis angle and the lateral deviation RMS. The exact levels of the parameters and the mean change from the level seat compared to the optimal correction are presented in Table 3.

In order to determine the effect of the inclination angle on the scoliosis angle and the lateral deviation RMS, the scoliosis angle and lateral deviation were not only considered with optimal correction but evaluated over all measurements. The results are presented in Figure 5, grouped according to severity and scoliosis type. Each point marks the magnitude of the respective parameter at the different inclinations. The inserted regression line shows the correlation between the effect of the angle of inclination and the type of scoliosis or the degree of severity. The steeper it is, the stronger the correlation. For all degrees of severity and scoliosis types, it could be seen that on average a more inclined wedge had a greater influence on the scoliosis angle and the lateral deviation RMS. If the elevation was on the ipsilateral side, the scoliosis angle decreased; if it was on the contralateral side, it increased. The *p*-value was always *p* < 0.05, except for the parameter lateral deviation RMS for thoracic scolioses (*p* = 0.302) and severe scoliosis (*p* = 0.173). Due to the statistical design of the study in relation to the main research question and the consequent lack of alpha correction of the secondary research question, these *p*-values do not demonstrate significance but should be understood as a trend. Nevertheless, it can be seen that the correlation between the effect of the inclination angle on the scoliosis angle/lateral deviation RMS and the type of scoliosis is greatest for lumbar scolioses (scoliosis angle: R^2^ = 0.313; lateral deviation RMS: R^2^ = 0.207). In terms of severity, mild scolioses also had a larger R^2^ value (scoliosis angle: R^2^ = 0.123; lateral deviation RMS: R^2^ = 0.047).

The recording of the subjective seat feeling using an NRS initially showed that an increase in the inclination is associated with a more uncomfortable feeling, regardless of which side is elevated by the wedge (Figure 6). When looking at the sensation of sitting on a level seat versus one with a wedge creating the optimal correction, it was found that the optimal correction was also more uncomfortable than the level seat. However, there were differences in the degree of worsening of the seating sensation for the different types and severities of scoliosis. The greatest deterioration in sitting sensation was seen in low scoliosis (up to 126%) and thoracic scoliosis (up to 131%) (Table 4).

## 4. Discussion

The present study investigated the influence of lateral seat wedges on the spine in patients with AIS. It is well known that the spine generally responds to seat inclinations in the sagittal plane with changes in the spine in the sagittal plane [26,27]. Changes in the frontal plane have so far been studied mainly in a standing position. There is a broad study base showing that unilateral pelvic elevation leads to a lateral curvature of the lumbar spine [15,28,29,30,31,32]. It has been observed that elevation of the shoe can have a positive effect on the spine [14,15]. However, Betsch et al. found that part of such a shoe lift ‘gets lost’ in small movements in the sacroiliac joint [29]. Significant changes in the spine occur only with a difference of >20 mm [33]. Such torsions possibly play a smaller role in the seat, as the transmission of the seat surface can be passed on more directly to the lumbar spine. For certain types of scoliosis, Lehnert-Schroth already recommends the use of a one-sided sitting elevation through, e.g., a sandbag on the lumbar convex side as part of Schroth therapy. However, it is also emphasized that this can lead to the formation or reinforcement of counter-curvatures [34].

In this study, it could be measured for the first time that in patients with AIS a lateral seat wedge with elevation on the ipsilateral side of the thoraco-lumbar/lumbar convexity leads more often to optimal correction of the rasterstereographic scoliosis angle (61%) as well as the lateral deviation RMS (67%) than a level seat or elevation on the contralateral side. The observed distribution for both parameters deviated significantly (*p* < 0.001) from an equal distribution. It should be mentioned that in some patients, a level seat or a contralateral elevation also led to optimal correction of the above-mentioned parameters. The position of a person’s spine therefore does not seem to be determined solely by the static component and does not always react in the same way to external influences such as an inclined seat. On the contrary, active posture is a factor that must be taken into account [35]. The therapy experience (for example how much experience someone has with active correction) as well as the subjective sitting sensation, the current mental state and sense of shame also play an important role [36,37,38].

Additionally, it was seen that different types and degrees of scoliosis showed different reactions to the seat wedges. Lumbar scolioses responded more often with optimal improvement to an ipsilateral elevation than did thoracic scolioses. So did low scolioses compared to severe ones. This was true for both parameters.

The magnitude of the correction also varied across the subgroups. Whereas the scoliosis angle showed a correction of about 23% in thoracic scolioses, it could be corrected by up to 44% with optimal inclination in lumbar scolioses. Comparing that to the acute ‘in-brace correction’ (IBC: radiologically measured Cobb angle after application of a brace), which is between 20 and 76% [2,39,40], the measured effect of lateral seat wedges for lumbar scoliosis seems to be numerically in a similar range. Knott et al. define at least 50% IBC to achieve ‘effective bracing’ [41]. It should be mentioned that IBC is measured radiologically by means of a spinal radiograph; in the present study, rasterstereographic measures were used.

The different results for the individual types of scoliosis and degrees of severity are probably due to the following: the spatial proximity of the pelvis to the lumbar spine presumably results in a more direct effect on lumbar curvature without attenuation by caudally located spinal segments. In addition, Hamazoglu et al. were able to show—congruent with the present results—that lumbar curves have greater flexibility and severe scolioses are stiffer [42]. Another cause could be the distribution of scoliosis types and severities. Thoracic scolioses had a higher mean Cobb angle in the present patient population. Since the effect of lateral seat wedges seems to be smaller in severe scolioses, this could explain—at least partially—the smaller correction in thoracic scolioses.

Considering and summarizing the above, AIS is a very heterogeneous pathology. The differentiation into scoliosis types and degrees of severity results in different reactions to laterally inclined seat wedges. Gram et al. emphasized in 1999 that idiopathic scoliosis is complex in its manifestation. When comparing the spines of patients with AIS in standing and sitting positions, different reactions to self-selected sitting and standing positions were found [43]. The knowledge of different patterns is incorporated in Schroth therapy, too. Distinct exercises are used according to the special Schroth patterns [34]. As a result, scoliosis-specific exercises have a better effect than normal physiotherapy and are recommended in the guidelines [2,44].

Another parameter recorded in this study was the subjective sitting sensation. Evidence was found that optimal correction of the scoliosis angle and lateral deviation RMS was not accompanied by an improvement in sitting comfort. However, the assessment revealed a different basis for evaluation. Some patients simply found it unfamiliar; for others it was a conscious evaluation in the sense of the ‘right’ or ‘wrong’ side of the elevation of the seat wedge due to previous therapy experience. It is possible that the patients need more time to adapt to the corrected posture and develop a new ‘normal’ and a better proprioceptive input.

There are limitations to this study. The first is in the patient population. There is a wide range for both the Cobb angle and the age. The inclusion of patients with a Cobb angle of 10–50° was chosen since this is the range for conservative treatment [45]. The age group was set at 10–18 including those aged 18, with the knowledge that this exceeds the limit for diagnosing adolescent scoliosis [2]. But boys tend to have a longer growing period than girls, with a fusion of the pelvic apophysis around the age of 18 [46]. Furthermore, the clearest effect for lumbar scoliosis could be shown only for a small subgroup within the total collective (n = 7). This is similar to the prevalence of scoliosis types in the study by Ponsetti et al. [47]. Differences from other distributions [48] are most likely to be seen in the definition of the scoliosis types in the classification. In the present study, the classification according to the current SOSORT Guidelines 2016 was used [2]. The results should be evaluated against the above background. However, it can be assumed that the trend shown will also persist in larger collectives or in those with a higher number of lumbar scolioses. Another limitation is the choice of parameters. Both parameters are measured in the frontal plane. The rasterstereographic scoliosis angle records the largest lateral curvature and does not allow any conclusions to be drawn about the entire spine. Any counter-curvatures that may arise are not recorded. In addition, due to the automatic detection of the largest curvature, the segments describing the angle may vary. The advantage of the rasterstereographic scoliosis angle, however, is that, despite the fact that it is ultimately measured in the frontal plane, it is based on a three-dimensional calculation of the vertebral bodies, and thus aspects of both the sagittal and transverse planes are included in this parameter [49]. In lateral deviation, the spine is recorded from VP (usually 7th cervical vertebra) to DM. The RMS is an averaged axial deviation of the entire spine in the frontal plane. In this study, the two parameters were used to detect the change in the spine when sitting on laterally inclined seat wedges, especially in the frontal plane, where the greatest effect was expected. Thirdly, the DL and DR landmarks were marked manually for this study. This was done despite the possibility of the automatic recognition of the landmarks due to problems with these during seated measurements. However, the advantage of better and constant intra-individual comparability of the measurements was considered greater than the potential sources of error during palpation [24] or soft tissue displacement [50,51]. The rasterstereographic measurements with the DIERS formetric 4D average proved to be a suitable method. Especially for measurement series, the advantage of being radiation-free should be emphasized. The fourth limitation is the use of wedges with three different degrees of inclination. The ‘optimal’ correction can thus only be considered optimal within these different tilts. Certainly, there is the possibility that an even better correction could be achieved by adding another seat inclination. Due to practical feasibility, the above-mentioned wedges with a slight, a medium and a strong inclination were chosen. Furthermore, only the acute reaction was measured. Statements about the direct effect during prolonged use or even a long-term effect cannot be made on the basis of the present study.

## 5. Conclusions

The study showed that optimal correction of the rasterstereographic parameters ‘scoliosis angle’ and ‘lateral deviation RMS’ by laterally inclined seat wedges is more often achieved by an ipsilateral elevation in relation to the thoraco-lumbar/lumbar convexity. In our view, the study has high clinical relevance. The possibility of having a positive effect on scoliosis while sitting is very high due to the long sitting time of approx. 9.7 h per day [16] and, if a positive influence on scoliosis emerges, it could bring a benefit for the patients concerned. Future studies could allow an even more comprehensive analysis of the effect of lateral seat wedges on the spine. It would make sense to focus on lumbar or thoraco-lumbar scoliosis, as this is where the greatest effect can be expected. In addition, it would be interesting to investigate whether comparable results can be achieved with longer use (e.g., the period of a school lesson) of such a seat wedge).

## Figures and Tables

**Figure 1 bioengineering-10-01086-f001:**
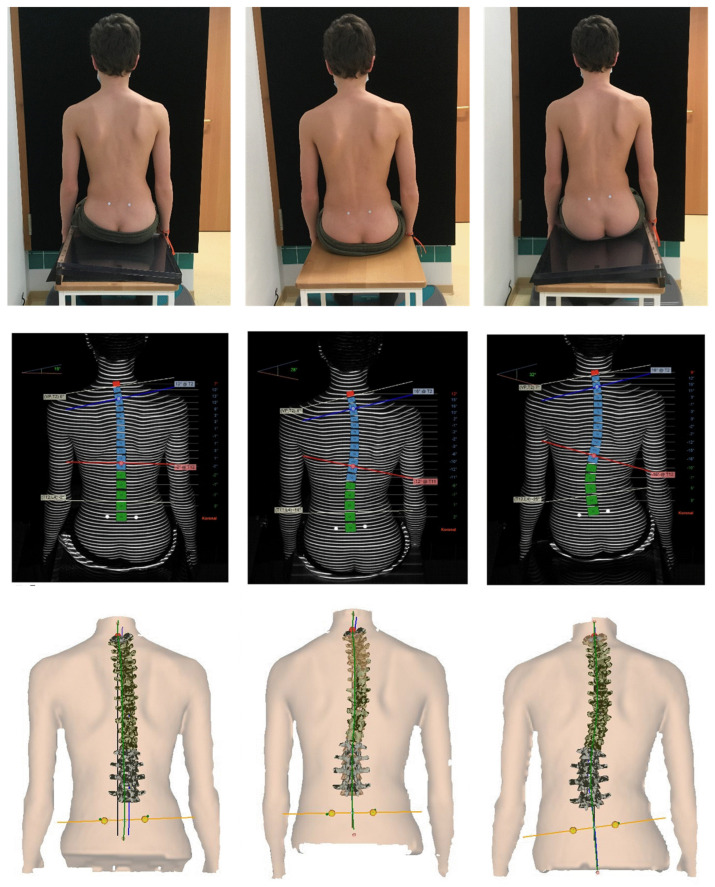
Patient with mild combined scoliosis; lumbar convexity on the left side. Top: test set-up with level seat surface (**centre**) and acrylic glass seat wedge with exemplary 6° inclination, elevation in each case on the ipsilateral (**left**) and contralateral (**right**) side of the lumbar convexity; lumbar dimples marked on the left and right side. Middle: representation of the distorted light grid on the patient’s back, calculated rasterstereographic scoliosis angle at the seat inclinations shown above. Blue line corresponds to the tangent of the cranial most inclined vertebra, red line corresponds to the tangent caudal most tilted vertebra; red vertebral body: VP; blue vertebral bodies: thoracic spine; green vertebral bodies: lumbar spine. Bottom: three-dimensional spine model calculated by DIERS DICAM; red dot corresponds to VP; yellow dots correspond to DL and DR.

**Figure 2 bioengineering-10-01086-f002:**
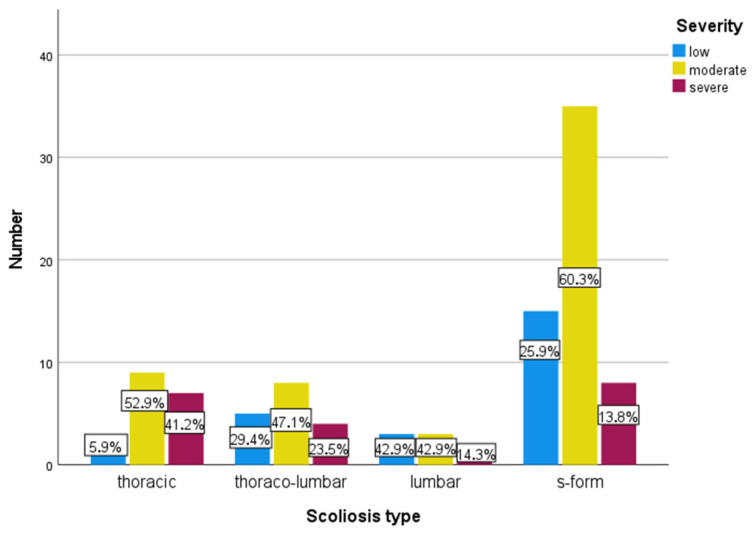
Frequency of severity for the different types of scoliosis; in the columns the relative shares are shown.

**Figure 3 bioengineering-10-01086-f003:**
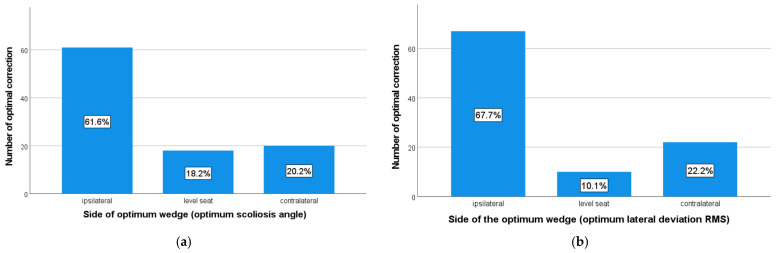
Frequencies of the side of the optimum wedge. The numbers in the columns show the relative frequency. (**a**) Optimum scoliosis angle. (**b**) Optimum lateral deviation RMS.

**Figure 4 bioengineering-10-01086-f004:**
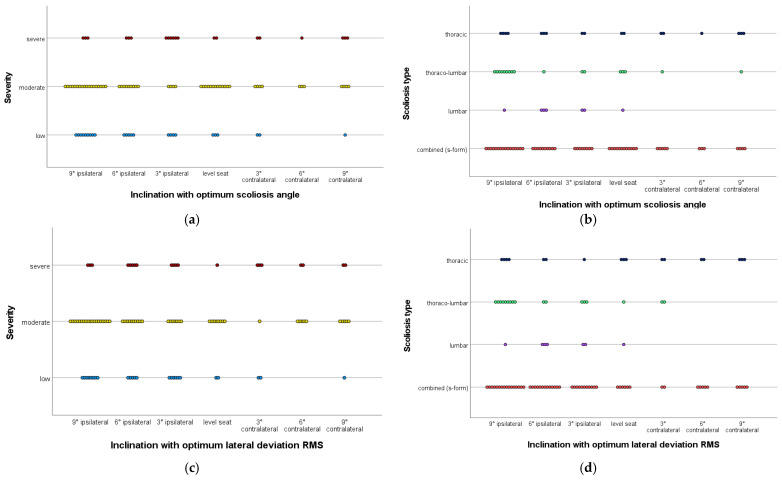
Frequency of the optimum correcting wedge as a function of severity and scoliosis type; each point marks a patient. (**a**) Optimum scoliosis angle, subgrouped according to severity. (**b**) Optimum scoliosis angle, subgrouped according to scoliosis type. (**c**) Optimum lateral deviation RMS, subgrouped according to severity. (**d**) Optimum lateral deviation RMS, subgrouped according to scoliosis type.

**Figure 5 bioengineering-10-01086-f005:**
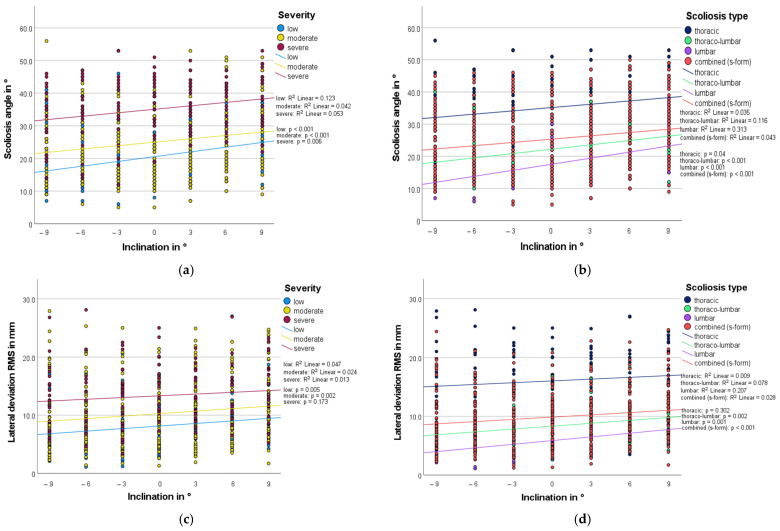
Scoliosis angle/lateral deviation RMS as a function of seat inclination. Negative sign equals ipsilateral; positive sign equals contralateral. Each point marks a patient; grouped regression lines are inserted. (**a**) Scoliosis angle, subgrouped by severity. (**b**) Scoliosis angle, subgrouped by scoliosis type. (**c**) Lateral deviation RMS, subgrouped by severity. (**d**) Lateral deviation RMS, subgrouped by scoliosis type.

**Figure 6 bioengineering-10-01086-f006:**
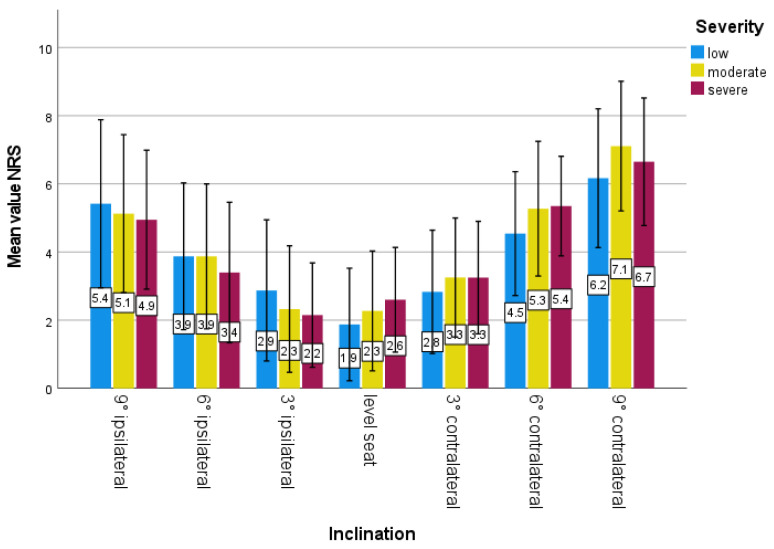
Mean NRS values as a function of seat inclination; grouped according to severity of scoliosis; error bars: ±1 SD.

**Table 1 bioengineering-10-01086-t001:** Height, weight, BMI, age in the total collective and subdivided by sex.

	Mean ± SD Total	Min/Max	Sex	Mean ± SD by Sex
size (cm)	167.1 ± 8.3	147/190	male	179.6 ± 8.8
			female	165.5 ± 6.8
weight (kg)	55.9 ± 9.6	38/84	male	63.1 ± 11.8
			female	55.0 ± 8.9
BMI (kg/m^2^)	20.0 ± 2.8	16.2/29.36	male	19.5 ± 3.4
			female	20.0 ± 2.7
age (years)	15.1 ± 1.6	11.7/18.8	male	16.0 ± 1.4
			female	15.0 ± 1.6

Mean ± standard deviation (SD); minima (min), maxima (max).

**Table 2 bioengineering-10-01086-t002:** Classification of scoliosis in the total collective.

Severity	Number (Percentage Share)	Scoliosis Type	Number (Percentage Share)
low	24 (24.2%)	thoracic	17 (17.2%)
moderate	55 (55.6%)	thoraco-lumbar	17 (17.2%)
severe	20 (20.2%)	lumbar	7 (7.1%)
		combined (s-form)	58 (58.6%)

**Table 3 bioengineering-10-01086-t003:** Averaged scoliosis angle (left) and averaged lateral deviation RMS (right) with respective standard deviations in level seat and with optimal correction of the respective parameter; in addition, the absolute and relative change is shown.

	Scoliosis Angle (°)			Lateral Deviation RMS (mm)		
	Level Seat	Optimal Correction	Change (Percentage Share)	Level Seat	Optimal Correction	Change (Percentage Share)
**Severity**						
low	19.3 ± 7.1	12.8 ± 5.2	6.5 (33.7%)	7.4 ± 3.4	4.7 ± 2.9	2.7 (36.5%)
moderate	24.4 ± 10.0	18.4 ± 8.3	6.0 (24.6%)	10.0 ± 5.2	7.0 ± 4.6	3.0 (30.0%)
severe	37.1 ± 8.5	27.5 ± 9.1	9.6 (25.9%)	14.1 ± 4.8	9.5 ± 4.3	4.6 (32.6%)
**Scoliosis type**						
thoracic	35.2 ± 11.7	27.2 ± 8.8	8.0 (22.7%)	15.7 ± 6.0	11.7 ± 5.8	4.0 (25.5%)
thoraco-lumbar	21.9 ± 7.2	15.2 ± 6.0	6.7 (30.6%)	8.0 ± 3.8	5.5 ± 2.6	2.5 (31.3%)
lumbar	18.6 ± 7.2	10.3 ± 3.4	8.3 (44.6%)	6.4 ± 4.1	3.1 ± 2.0	3.3 (51.6%)
combined (s-form)	24.9 ± 10.3	18.6 ± 8.9	6.3 (25.6%)	9.7 ± 4.3	6.4 ± 3.6	3.3 (34.0%)

**Table 4 bioengineering-10-01086-t004:** Mean NRS values on a level seat and with optimal correction of the scoliosis angle (left) and the lateral deviation RMS (right) with the respective standard deviation; in addition, the absolute and relative change are shown.

	Mean NRS Value				
	Level Seat	Optimal Correction of Scoliosis Angle	Change (Percentage Share)	Optimal Correction Lateral Deviation RMS	Change (Percentage Share)
**Severity**					
low	1.9 ± 1.6	4.2 ± 2.7	2.3 (121.1%)	4.3 ± 2.6	2.4 (126.3%)
moderate	2.3 ± 1.7	3.4 ± 2.3	1.1 (47.8%)	3.7 ± 2.4	1.4 (60.9%)
severe	2.6 ± 1.5	3.4 ± 2.6	0.8 (30.8%)	3.2 ± 2.2	0.6 (23.1%)
**Scoliosis type**					
thoracic	1.9 ± 1.2	4.4 ± 2.3	2.5 (131.6%)	4.0 ± 2.3	2.1 (110.5%)
thoraco-lumbar	1.8 ± 1.2	3.7 ± 2.2	1.9 (105.6%)	3.8 ± 2.1	2.0 (111.1%)
lumbar	1.9 ± 1.8	2.7 ± 2.1	0.8 (42.1%)	2.7 ± 2.1	0.8 (42.1%)
combined (s-form)	2.5 ± 1.8	3.4 ± 2.6	0.9 (36.0%)	3.8 ± 2.6	1.3 (52.0%)

## Data Availability

The data presented in this study are available on request from the corresponding author. The data are not publicly available due to privacy restrictions.

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
