# Peer review of "Influence of Lateral Sitting Wedges on the Rasterstereographically Measured Scoliosis Angle in Patients Aged 10–18 Years with Adolescent Idiopathic Scoliosis"

_bioengineering, 2023, doi:10.3390/bioengineering10091086_

Round 1

Reviewer 1 Report

This is an interesting study and the authors presented a unique subject in the follow up of and treatment of idiopathic adolecent scoliosis, with a practical and unique method. The paper is well written and structured. In my opinion the paper deserves to be published.  I hope the authors are goint to study succeeding studies in the future considering the lacking points they pointed out in the limitations section, such as  the 'wedge' desing and the differences of IAS and spinal and pelvic parameters in male and female.

Author Response

Dear Reviewer,

Thank you very much for your endorsement of our manuscript! We will consider the lacking points in future studies, and yes, we are currently preparing the next step, an intervention study with the measurement of short term effects of the wedges after about 15 minutes of sitting on it.

Kind regards,

Jürgen Konradi on behalf of the authors

Reviewer 2 Report

The title is clear and informative.

The abstract has all the necessary structural elements.

The study is easy to repeat following the Method chapter.

The topic is current. It is also very important.

I recommend publication in its current form.

Author Response

Dear Reviewer,

Thank you very much for your endorsement of our manuscript!

And for your interest, we are currently preparing the next step, an intervention study with the measurement of short term effects of the wedges after about 15 minutes of sitting on it.

Kind regards,

Jürgen Konradi on behalf of the authors

Reviewer 3 Report

Abstract: the text is well done either for content and style. It summarizes in effective way what Authors are describing in the full paper.

Introduction: it well describes what literature is reporting about this topic and progressively introduces the issue to the reader. Aim of the study is clearly reported and it is in accordance with the previous part of the introduction.

Methods: type of study is reported; Ethical approval is reported; Dimension of cohort in reported and inclusion/exclusion criteria are well described; Methodology for cohort studies is reported (STROBE); Methods of intervention are precisely reported, and primary vs secondary outcomes are well distinguished; the selected clinical scales have coherence with the aims of the study and scientific validity.

Statistics: well described and proportional to the aims of the study

Results: this paragraph shortly but precisely describes the results and Tables are presented in precise way in the aim to detail in a deeper way the results.

Discussion: it reports in adequate manner a concise but detailed analysis of results and describes clinical hypothesis to explain the results. Limitations of the study are reported.

Conclusions: no specific issues are related to this section.

Author Response

(The authors gave the same response as above.)
